# BoostStep: Boosting Mathematical Capability of Large Language Models via Step-Aligned In Context Learning

## Abstract

Large language models (LLMs) have demonstrated impressive ability in solving complex mathematical problems with multi-step reasoning and can be further enhanced with well-designed in-context learning (ICL) examples. However, this potential is often constrained by two major challenges in ICL: granularity mismatch and irrelevant information. We observe that while LLMs excel at decomposing mathematical problems, they often struggle with reasoning errors in fine-grained steps. Moreover, ICL examples retrieved at the question level may omit critical steps or even mislead the model with irrelevant details. To address this issue, we propose BoostStep, a method that enhances reasoning accuracy through step-aligned ICL, a novel mechanism that carefully aligns retrieved reference steps with the corresponding reasoning steps. Additionally, BoostStep incorporates an effective "first-try" strategy to retrieve for exemplars highly relevant to the current state of reasoning. BoostStep is a flexible and powerful method that integrates seamlessly with chain-of-thought (CoT) and tree search algorithms, refining both candidate selection and decision-making. Empirical results show that BoostStep improves GPT-4o's CoT performance by 4.6% across mathematical benchmarks, significantly surpassing traditional few-shot learning's 1.2%. Moreover, it can achieve an additional 7.5% gain combined with tree search. Surprisingly, it enhances state-of-the-art LLMs to solve challenging math problems using simpler examples. It improves DeepSeek-R1-671B and Qwen3-235B's performance on American Invitational Mathematics Examination (AIME) by 2.2% and 5.0% respectively, leveraging simple examples only from the MATH dataset.

## 1 Introduction

Mathematical reasoning is a crucial and challenging task in the development of artificial intelligence. It serves as an indicator of a model's ability to perform complex reasoning and has a wide range of applications, such as problem-solving, theorem proving, and scientific discovery.

When solving complex mathematical problems, cutting-edge LLMs often adopt a multi-step reasoning strategy. Specifically, they first decompose a complex problem into several simpler steps and then solve each single step independently.

Through the analysis of error cases, we found that current SOTA models are relatively correct in the step-dividing phase, that is, the model can know exactly what tasks should be completed in each step. However, there are still a lot of mistakes within each reasoning step, such as wrong formula use, wrong calculation, insufficient enumeration, etc. To quantitatively substantiate this observation, we provide GPT-4o-mini with a ground truth reasoning process to determine whether the error in another response was due to an overarching flawed reasoning approach or a deviation within a particular step. In less advanced models like LLaMA-3.1-8B (Dubey et al., 2024), 91.3% of errors originate from single-step reasoning. In more advanced models like GPT-4o, up to 99.2% of errors are ascribable to some particular steps. This exaggerated proportion suggests that the correctness of single-step reasoning is the bottleneck of reasoning capability.

Various approaches have been employed to improve reasoning correctness, such as producing chains of thought through prompt engineering (Kojima et al., 2022; Wei et al., 2022), fine-tuning with

mathematical data (Shao et al., 2024; Yang et al., 2024; Ying et al., 2024), or generating multiple candidate reasoning paths using Tree Search Methods (Zhang et al., 2024b;a; Wang et al., 2024b).

Among those techniques, in-context learning is a particularly important one, which offers similar examples to provide detailed guidance. However, the examples retrieved by traditional problem-level in-context learning are listed before the reasoning process, thereby lacking fine-grained guidance during the reasoning process. Moreover, since the example problem can't be identical to the new one, the irrelevant steps in those examples may even become a distraction from the current reasoning, thus even negatively affecting the single-step reasoning capability for some specific steps.

To this end, we refine in-context learning from problem-level to step-level granularity to offer similar example steps during an ongoing reasoning process for fine-grained step-aligned guidance. We also ensure that the introduced example is still relevant at the step level to avoid distractions.

Firstly, we have constructed an example problem bank with step-level granularity based on reasoning content instead of commonly adopted grammatical separation. This ensures the steps in the problem bank are consistent with the actual reasoning steps, thereby providing more appropriate guidance.

Building on the step-level granularity within the example problem bank, we propose an approach that incorporates in-context learning through a "first-try" format during an on-going reasoning process. For a given problem to be solved, we first break down the solving process into step-by-step reasoning paths. During the reasoning of a single step, we first allow the model to attempt a 'first try' to comprehend what the model currently needs to reason about. Based on this initial attempt, we then search the problem bank to find similar steps that can guide the model to accurately output the current step. This helps ensure a higher similarity between the retrieved examples and the current step so the distraction from irrelevant steps can be avoided and the guidance effect can be improved.

Compared with traditional problem-level ICL, our method provides examples during the reasoning process directly based on the steps to be solved, thereby offering more relevant guidance. It demonstrates significant improvements over traditional few-shot learning across various benchmarks, with an average increase of 3.4% on GPT-4o.

Moreover, our method also reduces the sensitivity to the similarity between the example and the target problem, as two different problems can still share similar steps. Consequently, dissimilar problems can still offer effective guidance. On multi-modal benchmarks with lower similarity to example problems, traditional few-host learning has a detrimental effect, resulting in an accuracy reduction of 0.9% on GPT-4o. In contrast, our approach still achieves an improvement of 2.8%.

Besides, BoostStep also shows a promising potential to improve the reasoning quality on harder problems with simpler examples. With examples from MATH (Hendrycks et al., 2021), it helps Deepseek-R1 and Qwen3-235B-Instruct-2507 (Yang et al., 2025) achieve an improvement of 2.2% and 5.0% respectively on the much more challenging American Invitational Mathematics Examination (AIME) problems.

Moreover, our method is also highly compatible with various current reasoning strategies that employ step-level tree search. Typically, a tree-search method requires a reason model to generate multiple step-level candidate reasoning paths and a critic model to evaluate the correctness of these candidates. Our approach can be integrated into both aspects. Specifically, when the reason model generates new candidate reasoning nodes, our method can introduce similar examples in the aforementioned 'first-try' manner to improve the accuracy of candidates. Additionally, it can aid the critic model by incorporating similar example steps into the evaluation of candidate reasoning processes to provide similar guidance. Experiments indicate that both applications contribute positively and bring about an improvement of 8.5% jointly on GPT-4o.

## 2 RELATED WORKS

**Mathematical Reasoning.** Mathematical reasoning has long been a challenging task in artificial intelligence. Early methods (Feigenbaum et al., 1963; Fletcher, 1985) attempted to perform simple mathematical reasoning through rule-based methods. With the advent of large language models with enhanced reasoning capabilities, contemporary approaches typically focus on enhancing performance during both the training and inference phases. The first category improves mathematical capability by fine-tuning with more high-quality mathematical data (Shao et al., 2024; Yang et al.,

2024; Lewkowycz et al., 2022; Yue et al., 2023; Xu et al., 2024). However, it demands substantial high-quality mathematical data and computational resources. Consequently, more efforts have been put into exploring various techniques during inference to enhance mathematical reasoning performance. Some work (Wei et al., 2022; Kojima et al., 2022) involves prompt engineering to enable models to generate comprehensive chains of thought. Others (Madaan et al., 2024; Gou et al., 2023; Ke et al., 2024) use self-refinement techniques to revise the initial reasoning outputs.

**Step-level Mathematical Reasoning.** Recently, many studies have shifted the granularity of mathematical reasoning from the problem level to the step level. This approach involves addressing each next step individually and completing small segments of reasoning within the overall task. These works often employ tree searching strategies like Tree of Thoughts (ToT) (Yao et al., 2024; Besta et al., 2024) or Monte Carlo Tree Search (Zhang et al., 2024b;a; Chen et al., 2024; Feng et al., 2023; Zhu et al., 2022), extending multiple steps to optimize step answers and ultimately obtain the optimal solution. Additionally, Process-Supervised Models (Lightman et al., 2023; Luo et al., 2024) are frequently used to verify the correctness of new candidate steps in real-time and prune reasoning paths. This more detailed auxiliary strategy demonstrates greater potential.

**ICL in Mathematical Reasoning.** In-context learning can provide low-cost guidance through similar examples. However, research on in-context learning within mathematical reasoning tasks remains insufficient. Typically, this approach involves providing the model with similar problems and their ground truth solutions to offer a general strategy for solving new problems (Hendrycks et al., 2021; Wei et al., 2022). Some efforts have been made to improve the relevance of retrieved examples by designing better retrieval mechanisms (Liu et al., 2024b). Others try to provide high-level context instead to improve the generalizability (Wu et al., 2024). Some recent approaches (Dong et al., 2024) introduce ICL into an on-going reasoning. However, all these methods share a common limitation: the lack of fine-grained step-level guidance. They still perform ICL in problem granularity and thus may not offer effective guidance for single-step reasoning.

## 3 STEP-LEVEL IN-CONTEXT LEARNING

### 3.1 REVISITING IN-CONTEXT LEARNING FROM CONDITIONAL PROBABILITY

Current models often employ next-token prediction for training and inference, where the conditional probability is central to the model's generation of the next token. Given a problem $q$, a model's reasoning process can be represented by $r_{predict} = \arg\max_r P_{model}(r \mid q)$, where we train the model to get a better conditional probability $P_{model}$ so that $r_{predict}$ can be closer to the ground truth answer $r_{gt} = \arg\max_r P_{gt}(r \mid q)$.

In-context learning provides the model with conditional probabilities similar to the ground truth answer for imitation without changing the probability model $P_{model}$. Specifically, an example problem $q'$ and its corresponding correct solution $r'$ is provided and it can be posited that the conditional probability $P(r' \mid q')$ is similar to the probability of the ground truth answer of the target problem $P(r_{gt} \mid q)$. Consequently, the model will imitate this similar example and $r'_{predict} = \arg\max_r P_{model}(r \mid q, q', r')$ will be closer to $r_{gt}$ comparing to $r_{predict}$.

However, given that the actual reasoning process $r$ can be highly complex, the complete reasoning process is often divided into multiple steps $s_1, s_2, \ldots$. Step-level reasoning iteratively guides the model to generate the next step $s_{i+1}^{0-shot} = \arg\max_s P_{model}(s \mid q, s_1, s_2, \ldots, s_i)$.

At the step granularity, examples retrieved based on the problem $q$ are evidently insufficient for providing appropriate guidance. Similar problem $q'$ may not necessarily contain the corresponding steps to guide the reasoning for the new problem $q$. Moreover, irrelevant steps may provide dissimilar conditional probabilities, thereby distracting the model's reasoning process.

To this end, we propose step-aligned in-context learning and a first-try strategy to provide detailed and relevant example steps when in step-level reasoning. Specifically, when generating new steps $s_{i+1}$ based on previous reasoning steps $s_i, s_{i-1}, \ldots, s_1$ and question $q$, we first utilize a first-try strategy to obtain an approximate estimate of $s_{i+1}^{first}$. Then, we use this $s_{i+1}^{first}$ to retrieve a similar step $s'_{n+1}$ along with the corresponding $q', s'_1, s'_2, \ldots, s'_n$. Since these two steps are similar, a very rea-

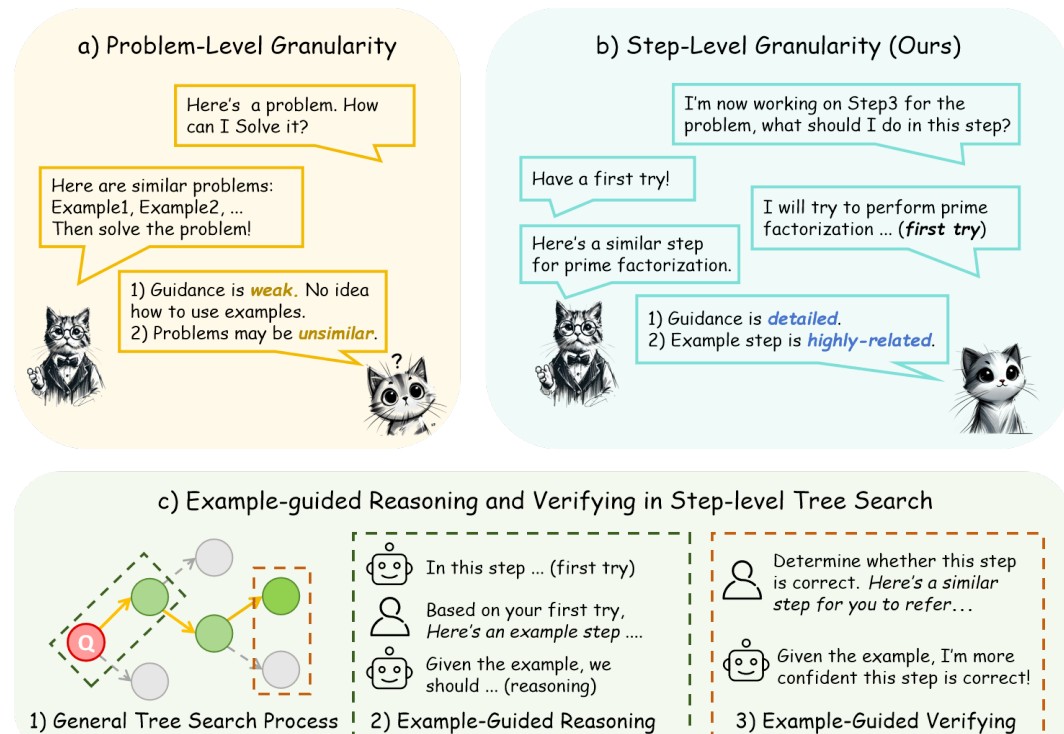

Figure 1: Our strategy refines in-context learning from problem-level granularity (fig.a) to step-level granularity(fig.b) to provide more real-time fine-grained guidance. Moreover, our strategy can guide the reasoning and verifying process in tree-searching strategies by introducing examples.

sonable assumption is that $P(s'_{n+1} \mid q', s'_1, \ldots, s'_n)$ closely approximates $P(s_{gt_{i+1}} \mid q, s_1, \ldots, s_i)$. Therefore, the generated step $s_{i+1} = \arg\max_s P_{model}(s \mid q, s_1, \ldots, s_i, q', s'_1, \ldots, s'_n, , s'_{n+1},)$ will be more closed to $s_{gt_{i+1}}$ comparing to $s_{i+1}^{0-shot}$. Details about our step-level in-context learning and first-try strategy will be explained in Sec. 3.3

### 3.2 STEP-LEVEL EXAMPLE PROBLEM BANK

Current open-source mathematical data no longer consist solely of problems and their final answers to determine whether the final answer obtained is correct or not. Instead, they also provide detailed solution processes to provide more fine-grained measurements. However, most current open-source mathematical data still do not break down the solution processes to the step level.

A major advantage of decomposing the question example bank into individual steps is that it facilitates step-level retrieval and guidance, which is of significant importance. As illustrated in Fig. 2, two distinctly different problems may contain similar key steps. Traditional problem-level in-context learning often overlooks such examples, whereas step-level in-context learning can effectively recall these steps, thereby providing fine-grained guidance to the ongoing reasoning process.

How to derive different steps from a complete solution is of great importance. Some approaches (Lightman et al., 2023) proposed using a clear semantic delimiter like the period '.' or a new line to segment steps. This allows for the quick decomposition without any additional assistance. However, this simple decomposition mode is obviously unreliable. A single reasoning step should have a consistent target and a complete thought process, making it the atomic granularity of reasoning. Using semantic delimiter may disrupt this atomicity. For example, it may split a complete enumeration for the same objective into multiple steps.

Therefore, we suggest that the most appropriate method for step segmentation is to allow the reason model itself to autonomously decompose the process. This approach ensures that the granularity of

> **Question:** The real number x satisfies 3x + \\frac{1}{2x} = 3. Find 64x^6 + \\frac{1}{729x^6}.
>
> **Step 1**: Multiplying both sides …
>
> **Step 2**: Squaring both sides, we get 4x^2 + \\frac{4}{3} + \\frac{1}{9x^2} = 4, so 4x^2 + \\frac{1}{9x^2} = \\frac{8}{3}
>
> **Step 3**: Cubing both sides, we …

> **Question:** Let x^8 + 98x^4 + 1 = p(x) q(x), where p(x) and q(x) are monic, non-constant polynomials with integer coefficients. Find p(1) + q(1).
>
> ……
>
> **Step 15**: If x - \\frac{1}{x} = 2 + i \\sqrt{6} then …
>
> **Step 16**: Squaring both sides, we get x^2 - 4x + 2 + \\frac{4}{x} + \\frac{1}{x^2} = -6, so x^2 - 4x + 8 + \\frac{4}{x} + \\frac{1}{x^2} = 0

Question Similarity: 0.01

Step Similarity: 0.81

Core Skills: When solving polynomials, it is often useful to consider expressions of the form (x + \frac{1}{x}) and square them.

Figure 2: Different problems may contain similar steps. Problem-level in-context learning will ignore this example due to low problem similarity. In contrast, our step-level in-context learning strategy can introduce the core skills by step-level retrieval and guidance.

the decomposed steps in example problem bank aligns with that of the real-time reasoning steps. Specifically, we define the concept of a step through prompts, which encapsulate a complete and simple inference. This guides GPT-4o in decomposing the answer at the step level.

We demonstrate some specific examples of different step-dividing strategies in Sec. 4.2.

### 3.3 STEP-LEVEL ICL WITH FIRST-TRY STRATEGY

The core challenge of in-context learning lies in how to effectively retrieve relevant problems or steps for effective guidance. This is contingent upon both the similarity between the problem database and the target problem, as well as the retrieval strategy employed. Traditional problem-level in-context learning involves retrieving similar problems based solely on the problem statement, which straightforward but effective, as similar problems typically encompass similar reasoning processes.

At the more granular step level, however, the situation becomes much more complex. A simple strategy is to perform retrieval using the given problem and all preceding reasoning steps $s_{i-1}, s_{i-2}, \ldots, s_1, q$. The clear drawback of this method is the excessive length of the retrieval content, which diminishes the emphasis on the uniqueness of the current step. Another strategy is to use the previous step $s_{i-1}$ to retrieve $s'_{j-1}$ from a step-level database, thereby guiding the reasoning of $s_i$ through the correct resolution of $s'_j$. However, this approach is rather crude, as it models step-level reasoning as a Markov process, which is evidently unreasonable. Similar steps can be applicable to different reasoning tasks, and therefore similarity in the previous step does not necessarily indicate that the retrieved subsequent step will provide valuable guidance for the reasoning in the current step.

To this end, we propose a straightforward and effective "first-try" strategy to enhance the similarity of search steps. Our premise is that the most accurate way to estimate the next step is to actually allow the model to attempt the reasoning for the next step. Specifically, given a problem $q$ and all preceding reasoning steps $s_{i-1}, s_{i-2}, \ldots, s_1$, we first instruct the model to attempt continuing the reasoning process to arrive at a tentative step $s_i^{try}$ without the aid of any examples. Subsequently, we use $s_i^{try}$ to retrieve similar steps $s'_j$ along with their corresponding problem $q'$ and preceding steps $s'_1, \ldots, s'_{j-1}$ from a step-level database. Finally, we feed the retrieved similar steps back to the model, enabling it to deduce the final step $s_i$. Besides, we add a widely accepted strategy reference rejection. Specifically, if the similarity of the retrieved most similar example remains below a certain threshold, we consider that there are no sufficiently similar examples available for reference and we do not provide any examples to avoid the negative effects associated with incoherent in-context learning. This "try-retrieve-reason" strategy significantly enhances retrieval relevance, thereby improving reasoning effectiveness. Experiments in Sec. 4.3 compare our method with several other retrieval strategies, demonstrating our superiority.

### 3.4 STEP-LEVEL GUIDANCE IN TREE SEARCH

Our step-level in-context learning can significantly enhance the model's single-step reasoning capability, which makes it easily integrated into common step-level tree-search strategies.

Table 1: **BoostStep generalizes across models and benchmarks.** Comparison of different ICL strategies on different benchmarks on GPT-4o, Qwen2.5-Math-72B-Instruct and Qwen3-32B.

| Model | Method | MATH | AMC12 | AMC10 | AQUA | MathBench(C) | MathBench(H) | Olympiad | **Avg** |
|---|---|---|---|---|---|---|---|---|---|
| GPT-4o | 0-shot | 73.4 | 53.6 | 55.8 | 81.1 | 80.0 | 77.3 | 40.6 | 66.0 |
| | few-shot | 73.8 | 56.5 | 56.7 | 83.9 | 80.7 | 79.3 | 39.3 | 67.2 (+1.2) |
| | **Ours** | **76.4** | **63.0** | **60.4** | **85.4** | **82.0** | **84.0** | **43.3** | **70.6 (+4.6)** |
| Qwen2.5 Math-72B | 0-shot | 83.0 | 67.4 | 67.7 | 84.6 | 80.6 | 82.0 | 49.7 | 73.6 |
| | few-shot | 83.8 | 67.4 | 66.8 | 85.0 | 81.3 | 82.7 | 49.9 | 73.8 (+0.2) |
| | **Ours** | **85.2** | **69.2** | **69.6** | **86.6** | **82.7** | **84.7** | **52.7** | **75.8 (+2.2)** |
| Qwen3-32B | 0-shot | 85.4 | 66.6 | 66.8 | 83.9 | 83.3 | 84.7 | 56.6 | 75.3 |
| | few-shot | 86.0 | 64.5 | 66.3 | 85.1 | 86.7 | 80.0 | 53.9 | 74.6 (-0.7) |
| | **Ours** | **87.6** | **68.9** | **69.1** | **85.1** | **90.7** | **87.3** | **57.0** | **78.0 (+2.7)** |

Generally, tree search methods necessitate two key components: a reason model that generates step-level reasoning and a Process-Supervised Reward Model (PRM) that continuously evaluates the current reasoning step in real time. Our method is beneficial for both of these components. It enhances the step-level reasoning performed by the reason model and improves the effectiveness of the PRM in evaluating current reasoning steps.

For the reason model, tree search methods inherently require step-by-step reasoning expansion. When expanding at node $s_i$, we can apply the previously mentioned strategy: the model performs $n$ first tries and retrieve for $n$ example steps. For each example, the model then completes the reasoning to generate $n$ child nodes $s_{i+1}^1, \ldots, s_{i+1}^n$ with the help of these examples. Similarly, our strategy can improve the accuracy of individual nodes $s_{i+1}^j$.

Evidently, judgment ability is closely related to reasoning ability. Therefore, since our strategy can enhance the accuracy of single-step reasoning, a reasonable assumption is that introducing appropriate example steps can improve the PRM's ability to assess the correctness of the current reasoning process. In particular, when evaluating the correctness of an inference step candidate $s_i^j$, we retrieve similar steps $s_k'$ along with their corresponding preceding steps $s_{k-1}', \ldots, s_1'$ and question $q'$ from the step-level example bank. Similarly, the probability distributions $P(s_k'|s_{k-1}', \ldots, s_1', q')$ and $P(s_{gt_i}|s_{i-1}, \ldots, s_1, q)$ exhibit similarities. This resemblance aids in assessing the discrepancy between $s_i^j$ and $s_{gt_i}$, thereby enhancing the accuracy of the critic model's evaluations.

Detailed ablation experiments in Sec. 4.4 demonstrate that both strategies contribute positivel.

# 4 EXPERIMENTS

## 4.1 EXPERIMENT SETTING

**Reasoning Model.** We conducted experiments on different reasoning models including GPT-4o (Hurst et al., 2024), which is our primary model, Qwen2.5-Math-72B-Instruct (Yang et al., 2024), Qwen3-32B (Yang et al., 2025) and SOTA rasoning models Qwen-QwQ-32B (Team, 2024), DeepSeek-R1-671B (Guo et al., 2025) and Qwen3-235B-A22B-Instruct-2507 (Yang et al., 2025).

**Evaluation Benchmark.** We tested our approach on several challenging open-source mathematical benchmarks. More details are listed in the appendix.

**Example Problem Bank.** The problems and the solutions are obtained from PRM800K (Lightman et al., 2023). Then we use our step-dividing strategy discussed above to divide example steps.

**Retriever.** We utilized the TF-IDF strategy as the retriever. The TF-IDF weight matrix is derived from the example problem bank because the impact of the newly generated step is negligible.

**Hyper-Parameters.** The temperature value is 0 in all the experiments except 0.3 at step-level tree search. The reference rejection threshold is 0.7. The shot number for traditional ICL is 4.

**Prompt.** The specific prompts are listed in the appendix.

Table 2: **BoostStep enables "simple-aids-complex"**. Simpler examples from PRM800K can guide LLMs on much more challenging AIME.

| Model | Method | AIME23 | AIME24 |
|---|---|---|---|
| QwQ-32B | 0-shot | 38.9 | 43.3 |
| | few-shot | 33.3 (-5.6) | 38.9 (-4.4) |
| | **Ours** | **41.1 (+2.2)** | **47.8 (+4.5)** |
| DeepSeek-R1 | 0-shot | 75.6 | 80.0 |
| | few-shot | 65.6 (-10.0) | 70.0 (-10.0) |
| | **Ours** | **77.8 (+2.2)** | **82.2 (+2.2)** |
| Qwen3-235B Instruct-2507 | 0-shot | 70.0 | 70.0 |
| | few-shot | 66.7 (-3.3) | 66.7 (-6.6) |
| | **Ours** | **73.3 (+3.3)** | **76.7 (+6.7)** |

Table 3: **BoostStep generalizes across modality.** Plain-text examples from PRM800K can provide effective guidance for MLLMs when solving multi-modal mathematical problems from MathVision and MathVerse, while traditional few-shot learning may even have negative impact.

| Method | MathVision | MathVerse |
|---|---|---|
| 0-shot | 30.6 | 53.2 |
| few-shot | 28.7 (-1.9) | 53.2 (0.0) |
| **Ours** | **35.2 (+4.6)** | **54.2 (+1.0)** |

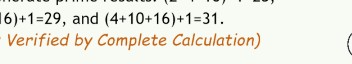

*Raw solution:*

First, we know that x is even. Every prime number larger than 3 is odd. So the sum of three members has to be even.
…
But 16 works because all the possible sums involving x=16 generate prime results. (2+4+16)+1=23, (2+10+16)+1=29, and (4+10+16)+1=31.
…

Grammatical Separation

Reasoning Content

Step 2: Every prime number larger than 3 is odd.
Step 3: So the sum of three members has to be even. *(Wrong Conclusion caused by Incomplete Content)*

Step 6: But 16 works because all the possible sums involving x=16 generate prime results. *(Unsupported Claims)*
Step 7: (2+4+16)+1=23, (2+10+16)+1=29, and (4+10+16)+1=31. *(Inexplicable Calculation)*

Step 2: Every prime number larger than 3 is odd. So the sum of three members has to be even. *(Correct and Complete Reasoning)*

Step 5:But 16 works because all the possible sums involving x=16 generate prime results. (2+4+16)+1=23, (2+10+16)+1=29, and (4+10+16)+1=31. *(Claims Verified by Complete Calculation)*

Figure 3: Our step division can provide complete and clear example steps as we divide the steps based on the reasoning content. By contrast, previous grammar-based dividing strategy may result in meaningless or even incorrect steps.

## 4.2 MAIN RESULTS

**BoostStep generalizes across models.** We evaluate BoostStep as a model-agnostic strategy across both general-purpose and math-specialized LLMs. As shown in Tab. 1, BoostStep brings consistent improvements to both GPT-4o and Qwen-Math, demonstrating its broad applicability. The performance gains across all benchmarks confirm that BoostStep does not rely on model-specific tuning or architectural assumptions, but instead offers universally beneficial step-level guidance.

**BoostStep generalizes to out-of-bank benchmarks.** Notably, the example bank used by BoostStep is constructed solely from the PRM800K training set (i.e., MATH500), yet the method achieves strong performance across a wide range of math benchmarks. This out-of-distribution generalization highlights the robustness of the step-level design: even when benchmark distributions differ, BoostStep increases the likelihood of identifying transferable intermediate steps, enabling effective reasoning in unfamiliar problem domains.

**BoostStep enables "simple-aids-complex".** As shown in Tab. 1 and Tab. 2, although the example bank is constructed from PRM800K—an easier dataset than the evaluated benchmarks—the step-level design still provides valuable guidance for solving specific steps within complex math problems. This demonstrates that even a relatively simple bank can effectively enhance model performance on more challenging tasks by offering transferable step-wise insights.

**SOTA reasoning LLMs also benefit from BoostStep.** While SOTA reasoning LLMs already exhibit strong performance on complex mathematical problems, BoostStep can still bring gains by providing accurate step-level guidance, as evidenced in Tab. 2. In contrast, traditional few-shot learning often fails to deliver effective support, highlighting the unique advantages of BoostStep's structured, step-wise assistance even for advanced models.

**BoostStep generalizes across modality.** Despite differences in input modality, multi-modal mathematical reasoning still follows a step-by-step logical process. BoostStep leverages this shared

Table 4: Comparing Booststep with some step-level reasoning strategies without In-context learning.

| Method | MATH | AMC12 | AMC10 | Avg |
|---|---|---|---|---|
| ToT | 77.8 | 58.7 | 59.0 | 65.17 |
| Self-Refine | 73.0 | 51.4 | 54.8 | 59.73 |
| **Ours** | **76.4** | **63.0** | **60.4** | **66.60** |

Table 5: Comparing BoostStep with other advanced ICL methods.

| Method | MATH |
|---|---|
| IDS | 74.2 |
| LMS3 | 75.2 |
| **Ours** | **76.4** |

Table 6: **Robustness toward bank quality.** Experiments on the sensitivity of the similarity between questions and examples. R_t indicates that the examples are the t_th similar without any rejection.

| Method | Math-level5 | AMC12 | AMC10 |
|---|---|---|---|
| 0-shot | 50.7 | 53.6 | 55.8 |
| few-shot R_1 | 52.2 (+1.5) | 56.5 (+2.9) | 56.7 (+0.9) |
| few-shot R_4 | 46.3 (-4.4) | 52.2 (-1.4) | 53.7 (-2.1) |
| Ours R_1 | 56.0 (+5.3) | 62.3 (+8.7) | 60.4 (+4.6) |
| Ours R_4 | 52.2 (+1.5) | 61.6 (+8.0) | 58.1 (+2.3) |

Table 7: **Effectiveness of our First-Try Strategy.** We compare our strategy with two other step-level retrieval strategies. 'Path' represents using the complete reasoning path $s_{i-1}, \ldots, s_1$ while 'Pre-Step' represents using the previous step $s_{i-1}$.

| Strategy | AMC12 | AMC10 | MATH |
|---|---|---|---|
| Path | 56.5 | 58.1 | 73.8 |
| Pre-Step | 57.2 | 56.7 | 74.0 |
| **First-try** | **63.0** | **60.4** | **76.4** |

structure to provide effective guidance when applied to multi-modal benchmarks. We evaluate our method on MathVision (Wang et al., 2024a) and MathVerse (Zhang et al., 2025), using the plaintext examples from PRM800K. As shown in Tab. 3, traditional few-shot prompting at the problem level not only fails to improve performance but may even degrade reasoning quality. In contrast, BoostStep consistently delivers substantial gains, demonstrating its ability to bridge modality gaps and support systematic reasoning under more complex, visually grounded scenarios.

**BoostStep outperforms other advanced strategies.** We compare our BoostStep with step-level reasoning strategies without ICL in Tab. 4 and other advanced ICL strategies in Tab. 5. For step-level reasoning without ICL, we follow the settings in our paper and reproduced the performance of the GPT-4o model on Tree of Thoughts (Yao et al., 2024) and Step-level Self-Refine (Madaan et al., 2024) methods across the AMC10, AMC12, and MATH500 test sets. For other advanced ICL methods, since these methods are not open-sourced, we are unable to reproduce and compare them on a broader set of test datasets. Therefore, we compare our results on the MATH dataset with the accuracy reported in the LSM3 (Liu et al., 2024b) and IDS (Qin et al., 2023) papers.

### 4.3 ANALYSIS ON BOOSTSTEP

**Robustness toward bank quality.** The sensitivity toward the example bank quality is a fundamental challenge for ICL, and BoostStep significantly mitigates this sensitivity by providing step-level guidance. Here we study its robustness quantitatively. As shown in Tab. 6, we decrease the similarity by selecting the t_th similar example during reasoning. We observe that traditional ICL suffers from a severe decrease and is even worse than 0-shot learning when t is larger than 4. In contrast, our method does not show a significant decline and is consistently better than the 0-shot reasoning.

**Effectiveness of reasoning-based step division.** To better align with the steps in reasoning, we propose constructing a step-level problem bank based on the reasoning content rather than grammatical divisions. To prove our assumption, we compare our approach with a commonly used strategy that constructs steps based on grammatical segmentation, using periods '.' as the delimiter, on the same dataset PRM800K and under identical conditions. Results are presented in Tab. 8. Our method largely outperforms those using periods as a delimiter. Fig. 3 demonstrates a specific example of different step-dividing strategies. Dividing by grammatical separation may break a complete reasoning step into several incomplete fragments, thereby losing its guiding value. While ours can provide complete and clear example steps.

**Effectiveness of First-Try Strategy.** The key factor of in-context learning lies in the relevance of the retrieved examples. At the finer-grained step level, designing an appropriate retrieval strategy becomes even more crucial and challenging. Therefore, we propose the first-try strategy, which involves understanding what the model currently needs to reason about using a first attempt and then searching the problem set for similar steps to guide the model in fully outputting the current step. To validate the effectiveness of this method, we compare it with several other strategies mentioned

Table 8: Comparison of different step-level example problem Bank construction methods. 'GS' represents using Grammatical Separation '.' as delimiter while our strategy use the reasoning content to divide the steps.

| Strategy | AMC12 | AMC10 | MATH |
|---|---|---|---|
| GS | 56.5 | 58.1 | 74.8 |
| **Ours** | **63.0** | **60.4** | **76.4** |

Table 9: Detailed ablation on incorporating retrieving similar example steps during the reasoning and verifying phases of step-level tree search methods.

| Reason | Verify | AMC12 | AMC10 | MATH | **Avg** |
|---|---|---|---|---|---|
| w/o tree-search | | 53.6 | 55.8 | 73.4 | 60.9 |
| ✗ | ✗ | 58.7 | 59.0 | 77.8 | 65.2 (+4.3) |
| ✓ | ✗ | 64.4 | 62.2 | 79.2 | 68.6 (+7.7) |
| ✗ | ✓ | 61.6 | 60.4 | 78.2 | 66.7 (+5.8) |
| ✓ | ✓ | **65.2** | **63.6** | **79.4** | **69.4 (+8.5)** |

in Sec.3.3, retrieving by the entire reasoning path $s_{i-1}, s_{i-2}, \ldots, s_1, q$ or only by the immediately preceding step $s_{i-1}$. Tab. 7 presents the detailed result. Our method significantly outperforms the other two strategies, better anticipating the content that needs to be inferred in the current step.

**Efficiency.** To provide appropriate examples at the step granularity, we have introduced more sophisticated reasoning and retrieval mechanisms. Therefore, the additional time cost also warrants discussion. The extra cost is primarily attributable to the per-step retrieval and the first-try strategy. Fortunately, owing to the adoption of appropriate strategies, neither introduces significant time costs.

For example retrieval, since the TF-IDF vectors of the example bank can be precomputed, what needs to be encoded and computed during real-time reasoning is actually minimal, resulting in a negligible time cost. Quantitatively, a single retrieval takes only a few milliseconds, which accounts for less than *1%* of the time required by any model.

For the first-try part, a rejection strategy is adopted: if the similarity of the most similar example step is still below a certain threshold, we directly use the first-try as the inference content. This strategy ensures the quality of the provided examples while also improving the efficiency of our approach. Quantitively, the first-try attempt will only add *30%* time cost to our reasoning.

**Case Study.** We provide a specific example of how Booststep improves single-step reasoning through example steps in Sec. C in the appendix.

## 4.4 EXTENDING BOOSTSTEP TO TREE SEARCH

The reasoning capability of the reason model and the verifying capability of the critic model are two core factors of step-level tree search methods, and our strategy can bring benefits in both ways. On one hand, it can improve the accuracy of generating candidate nodes using the previously mentioned first-try strategy when reasoning nodes are generated. On the other hand, it can increase the accuracy of evaluation by introducing similar examples during critic model assessments and therefore ensures that the correct reasoning nodes are more likely to be preserved. These can be decoupled, allowing us to demonstrate the effectiveness of each component through ablation studies.

We utilize GPT-4o as the reason model, GPT-4o-mini as the PRM and adopt the Pairwise Preference Reward Model (PPRM) configuration (Zhang et al., 2024b). Detailed settings of our tree search method will be listed in the appendix.

Tab. 9 presents the results of integrating in-context learning into the reasoning and evaluation phases of Tree Search methods. The results of this ablation study indicate that introducing example steps can enhance both the reasoning and verifying capabilities of tree search methods. Therefore both approaches contribute to the improvement of overall reasoning performance.

## 5 CONCLUSION

We propose BoostStep, addressing two criticial challenges in previous In-Context Learning strategies: granularity mismatch and irrelevant information. BoostStep can provide highly-related examples at the step granularity, thereby providing fine-grained guidance during an on-going reasoning. BoostStep is a strong and general approach which can enhance the model's reasoning capabilities and reduce the sensitivity of the examples. It can break through the limitations of traditional ICL like achieving 'simple-aids-complex' and cross-modal guidance. Moreover, it can be applied in tree search methods to enhance the reasoning and verifying capability.

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

# A    DETAILED EXPERIMENT SETTING

## A.1    PROMPT

**Prompt for 0-shot COT**: You are a professional math problem solver. Solve the problem step by step and output the final answer within \\boxed{}.

**Prompt for problem-level few-shot learning**: You are a professional math problem solver. Solve the problem step by step and output the final answer within \\boxed{}. In case you don't know how to solve it, I will give you example problems with their full solutions which you can refer to.

Example i:

Problem: xxx

Solution: xxx

**Prompt for first-try in step-level COT**: You are a professional math problem solver. I will give you a math problem and part of its solution. And you need to only output the next step of the solution, starting with 'Step $i$:', where $i$ is the step number. If you think that the final step is derived, put the answer within \\boxed{}.

**Prompt for step-level few-shot learning**: You are a professional math problem solver. I will give you a math problem and part of its solution. And you need to only output the next step of the solution, starting with 'Step $i$:', where $i$ is the step number. In case you don't know how to derive the correct content, an example with 'Key Step' will be given. You need to learn how 'Key Step' is derived, and implement similar strategy in your derivation procedure. If you think that the final step is derived, put the answer within \\boxed{}.

Example Problem: xxx

Example Solution: Step 1: xxx, Step 2: xxx, ..., Step i (Key Step): xxx.

## A.2    DETAILS OF GRADING AND METRICS

We follow the setting of Opencompass (Contributors, 2023) and VLMEvalKit (Duan et al., 2024). Specifically, we first require the model to put the final answer within \\boxed{}. Then, we use GPT-4o-mini as the judge model to compare the final answer with the ground truth answer. Compared to string matching, this approach can eliminate some false negative evaluations because the same mathematical expression can be expressed in many forms. If the model fails to follow the the expected format in the prompt and the rule-based extraction fails, the solution is directly judged as inconsistent with ground truth.

## A.3    BENCHMARKS

We tested our approach on several mathematical benchmarks, including MATH500 (Hendrycks et al., 2021), AQuA (Ling et al., 2017), OlympiadBench-TO (He et al., 2024) and MATHBench (Liu et al., 2024a) and the real questions of American Invitational Mathematics Examination (AIME) in year 2023 and 2024. Specifically, we use the Olympiad-TO (text-only) subset of OlympiadBench and the application problems in college-level and high-level difficulty of MATHBench.

In addition, we collected problems from the AMC-10 and AMC-12 competitions to serve as more challenging benchmarks. They include real questions from the 2018-2023 AMC exams.

For multi-modal math benchmarks, we use MathVision-Mini (Wang et al., 2024a) and vision-dominant version of problems in MathVerse-Mini (Zhang et al., 2025).

***Question:*** For some value of x, 0 < x < 180, tan 53° tan 81° tan x° = tan 53° + tan 81° + tan x°. Find x.

***First-Try:*** Use the identity for the tangent of the sum of angles, tan (A + B + C) = tan A + tan B + tan C - tan A tan B tan C *(Wrong Equation)* and compare it with the given equation. 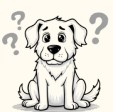

***Example Step:*** From the addition formula for tangent, tan (A + B + C) = \frac{tan A + tan B + tan C - tan A tan B tan C}{1 - (tan A tan B + tan A tan C + tan B tan C)} *(Key Example Equation)*. Since A + B + C = 180°, this is 0. Hence, tan A + tan B + tan C = tan A tan B tan C. 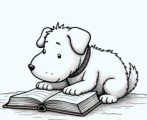

***Final-Reasoning:*** Notice that the equation resembles the identity for the tangent of a sum of angles. We can use the identity: tan(A + B + C) = \frac{tan A + tan B + tan C - tan A tan B tan C}{1 - (tan A tan B + tan B tan C + tan C tan A)} *(Correct Equation)*

***Next step:*** Since the equation is symmetric in terms of the tangents, we can assume that the angles 53°, 81°, and x° sum to 180°. Therefore, we have: **53° + 81° + x° = 180°.** *(Correct Conclusion)*

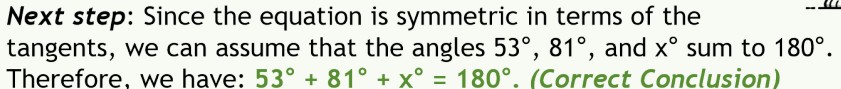

Figure 4: A specific example of adjusting reasoning during real-time inference through step-level in-context learning. The first try uses a wrong equation while the retrieving example step guides the model to use the correct equation and get the correct conclusion.

## B    DETAILED SETUP FOR EXAMPLE-GUIDED STEP-LEVEL TREE SEARCH

In the setup for tree search methods, we utilize GPT-4o as the reason model and employ GPT-4o-mini as the Process-supervised Reward Model (PRM). For the PRM, we adopted the Pairwise Preference Reward Model (PPRM) configuration (Zhang et al., 2024b). Specifically, PPRM transforms the absolute rewards calculation into preference predictions between solutions to calculate rewards. This approach reduces the variability associated with scoring characteristics and thus leads to a more robust and consistent evaluation of different solutions.

The complete reasoning process in our experiment is as follows: we start with the target problem as the root node and obtain two initial solution steps through sampling to serve as the two initial parent nodes. In each step-level reasoning phase, we expand these two parent nodes through sampling, generating four candidate child nodes. Using the PPRM, we select the two child nodes with higher confidence to become the parent nodes for the next step of reasoning. This process continues until both candidate nodes have completed their reasoning paths, resulting in the final answers. Finally, PPRM is used to select the ultimate answer from these two reasoning paths.

## C    CASE STUDY

Here we demonstrate a specific example of how our step-level in-context learning boosts step-level reasoning. Given the question, we first let the model have a first try on step one. Unfortunately, because the model is unfamiliar with trigonometric functions, it makes an error on the tangent sum formula, therefore leading to a wrong step. However, we can get a rough idea of what the model wants to calculate at this step according to the first try. Then, we find a similar step that correctly leverages the tangent sum formula in the step-level example problem bank. Therefore, with the guidance provided, the model correctly applied the tangent sum formula during the second reasoning attempt and arrived at the correct answer.

