# OpenReview forum: "BoostStep: Boosting Mathematical Capability of Large Language Models via Step-aligned In Context Learning"
_ICLR.cc/2026/Conference — Submitted to ICLR 2026_

### Official Review · Reviewer_GjNr · 2025-10-23

**Soundness:** 3
**Presentation:** 3
**Contribution:** 3
**Rating:** 6
**Confidence:** 3

**Summary:**

This paper introduces BoostStep, a novel method to enhance the mathematical reasoning capabilities of LLMs by refining in-context learning (ICL). BoostStep addresses the issues of traditional ICL by shifting ICL from the problem level to the step level. The core of the method is a "first-try" strategy: for each step in a problem, the model first generates an initial attempt. This attempt serves as a semantic query to retrieve a highly relevant, correct reasoning step from a pre-compiled example bank. This targeted example then guides the model to produce a more accurate final step. The paper also emphasizes the importance of creating the example bank by segmenting solutions based on reasoning content rather than simple grammatical delimiters.

The authors evaluate across multiple models (GPT-4o and Qwen variants) and mathematical benchmarks (MATH, AIME, MathVerse). The results consistently show that BoostStep outperforms both zero-shot and traditional few-shot ICL baselines. Notably, the method demonstrates the ability to use simpler examples to solve more complex problems, generalizes to out-of-distribution benchmarks, and even provides gains in multi-modal reasoning tasks. Finally, the paper shows that BoostStep can be effectively integrated into tree-search algorithms, improving the performance of both the reasoning and verification components.

**Strengths:**

*   **Observation and Problem Formulation:** The paper identifies a fundamental bottleneck in LLM reasoning—that errors are often local to a specific reasoning step, while guidance from traditional ICL is global. This reframing of the problem from problem-level to step-level is both intuitive and powerful.
*   **Effective Method:** The proposed "first-try" retrieval mechanism is an effective solution. Using the model's initial reasoning attempt as a rich semantic query to find a relevant example is a significant improvement over methods that rely only on previous correct steps or the overall problem statement.
*   **Comprehensive Evaluation:** The experimental validation is thorough and convincing. The authors use multiple state-of-the-art models and mathematical benchmarks. The performance gains reported in Tables 1, 2, and 3 strongly support the paper's claims.
*   **Generalization:** The method shows remarkable robustness. The "simple-aids-complex" result (Table 2), where examples from the MATH dataset help solve more challenging AIME problems, is particularly impressive. Furthermore, its ability to improve performance on out-of-distribution and even cross-modality benchmarks (Table 3) highlights that BoostStep learns transferable reasoning patterns.

**Weaknesses:**

*   **Dependence on Example Bank Quality:** The performance of BoostStep is tied to the quality and coverage of the step-level example bank. The paper notes strong performance using a bank built from PRM800K, but it would be beneficial to include a brief discussion on the method's sensitivity to the bank's size and diversity.
*   **Details on Step Segmentation:** The authors propose a superior method for dividing solutions into steps based on "reasoning content" rather than grammatical delimiters. This is a key part of the contribution. While Figure 3 shows a compelling example, the paper could benefit from slightly more detail on the specific prompts or methods used to guide the LLM in performing this crucial segmentation task during the creation of the example bank.

**Questions:**

* **Failure Analysis:** Could you provide insight into the primary failure modes of BoostStep? For example, are there cases where a retrieved step, despite its high similarity score, actually misleads the model due to a subtle contextual difference between the two problems?

* **Sensitivity to the Rejection Threshold:** The paper uses a similarity threshold of 0.7 to decide whether to provide an example. How was this value determined, and how sensitive is the overall performance to this hyperparameter?

* **Applicability to Other Domains:** While the method is brilliantly applied to mathematical reasoning, its core idea seems applicable to other domains requiring complex, sequential reasoning (e.g., programmatic code generation, legal reasoning, or complex question answering). Have you considered the potential of BoostStep in these other areas?

* **Dynamic Knowledge Integration:** The example bank is constructed offline. Do you see a path forward for a system where BoostStep could dynamically integrate newly solved and verified reasoning steps into its own bank, creating a self-improving reasoning system over time?

---

### Official Review · Reviewer_DQnD · 2025-10-25

**Soundness:** 2
**Presentation:** 2
**Contribution:** 1
**Rating:** 2
**Confidence:** 5

**Summary:**

This paper introduces BoostStep, a step-aligned in-context learning framework for mathematical reasoning with LLMs. The method builds a bank of step-level exemplars by segmenting worked solutions, runs a quick first-try to guess the next step for the current problem, retrieves similar steps from the bank, and then conditions the model on those exemplars to generate and verify the next step. The authors also plug this idea into tree-style search with a process-reward model, injecting examples during both generation and verification. On several math benchmarks they report gains over zero-shot and standard few-shot prompting, and provide ablations on segmentation and retrieval choices. While the high-level idea is practical and intuitive, I find the empirical support incomplete and the evaluation not aligned with current  standards.

**Strengths:**

The paper targets a real pain point. i.e. problem-level exemplars often do not align with the local reasoning step. The proposed pipeline is straightforward to implement, integrates seamlessly with tree search, and appears to consistently produce improvements over plain few-shot learning on several datasets. I also appreciate the attempt to improve step segmentation beyond naive punctuation splitting, which is a practical detail many works gloss over.

**Weaknesses:**

The most serious issue is the already outdated and selective benchmarking. The idea of aligning ICL at the step granularity with a first-try cue is neat from an engineering perspective, but the novelty is incremental compared to existing step-wise reasoning and retrieval-augmented prompting.

More importantly, the contribution is weakened by the evaluation choices as the model set is not up-to-date and coverage across benchmarks is uneven, so it is hard to judge how competitive this really currently is, especially when considering the fast-paced development. Newer frontier models are only represented in a narrow AIME experiment and not across the full suite. This does not meet ICLR2026 quality expectations and makes the gains hard to interpret.

On methodology, retrieval is under-analyzed (no quality metrics, no dense/hybrid baselines), statistics are missing (no CIs, no seed variation), and the grading depends on a related model family instead of an independent checker or human audit. Efficiency is not convincingly quantified, given the extra pass and longer prompts. I also worry about potential leakage at the step level since the bank is built from overlapping sources. There is no near-duplicate analysis to rule this out. Finally, several ablations are incomplete (no sensitivity to the rejection threshold, no study of exemplar count and no taxonomy of failure cases where the retrieved step actually misleads).

**Questions:**

1) Will you re-run the current frontier models (e.g., the latest math-specialized open and strong closed models) across all main benchmarks and report compute-matched numbers with confidence intervals?

2) Can you provide retrieval diagnostics (precision@k/nDCG) and compare TF-IDF with a modern dense retriever and a hybrid setup, including a sensitivity study for the rejection threshold and exemplar count k?

3) How robust are your results to the judge? Please re-score a stratified subset with exact/symbolic matching, a different model family as grader, and, where feasible, human adjudication.

4) Please include end-to-end cost/latency per problem (tokens and wall-clock) for base few-shot vs. BoostStep, both with and without tree search.

5) Did you run a near-duplicate audit at the step level between the step bank and each test set? If yes, please quantify.

---

### Official Review · Reviewer_xqRf · 2025-10-29

**Soundness:** 2
**Presentation:** 3
**Contribution:** 2
**Rating:** 4
**Confidence:** 4

**Summary:**

The paper proposes BoostStep, which aims to enhance reasoning accuracy through aligning retrieved reference steps with the corresponding reasoning steps. It introduces a first-try strategy to ensure high relevance between retrieved examples and current reasoning, marking a shift from problem-level to step-level ICL. Booststep can be combined with CoT and tree search algorithms to improve LLMs to solve math problems.

**Strengths:**

- BoostStep refines in-context learning from the problem level to the step level, enabling guidance for each reasoning step rather than coarse whole-problem imitation.
- It introduces a novel first-try strategy: the model first attempts a reasoning step, then retrieves the most similar example step. This improves relevance and reduces distraction from irrelevant examples.
- The method can be integrated into CoT and tree-search frameworks, which further enhances the model’s performance.

**Weaknesses:**

- When the model’s initial reasoning output is poor, the system cannot effectively perform step-wise in-context learning, since the retrieval stage depends on the content of this initial attempt. In such cases, BoostStep either retrieves irrelevant examples or rejects retrieval altogether, providing no additional guidance.
- The accuracy of LLM’s automatic step-splitting is not guaranteed. The construction of the step-level problem bank needs further validation.
- The first-try → retrieve → re-reason cycle adds 30% more inference time, which may become too expensive and less practical for large-scale applications. The trade-off should be further investigated and addressed.

**Questions:**

- As the performance of BoostStep depends on retrieving semantically similar step-level examples, how would the method scale with a significantly larger or more diverse step-level problem bank?
- Could the authors provide the experimental results for BoostStep when used without the MCTS component? It would be helpful to isolate the contribution of step-aligned in-context learning alone.
- Can the step-aligned ICL framework transfer to domains beyond mathematics, such as code reasoning or physics derivations, without re-collecting a new step-level example bank?

---

### Official Review · Reviewer_JDZW · 2025-11-01

**Soundness:** 2
**Presentation:** 1
**Contribution:** 2
**Rating:** 2
**Confidence:** 4

**Summary:**

The paper describes a method to perform stepwise retrieval method for ICL called BoostStep, which performs a draft reasoning for retrieval, gain improvements over a few shot baseline on a collection of datasets.

**Strengths:**

1. improvement over baseline few shot ICL method
2. nice combination with tree-based searching methods

**Weaknesses:**

1. missing a couple of details in the experiments and claims
2. the comparison baseline is crispy, similar methods such as IDS (Qin et al. EMNLP findings 24) and LMS3 (Liu et al. 24) are only compared on one dataset, and the base model is not even the same, the reported results for comparison was conducted on GPT-4 while the major results were conducted on GPT-4o, no evidence shows that the current method can outperform these two baselines
3. the ablation of the method itself is also crispy, see questions part for details. The current experiments does not reveal where the performance gains from

**Questions:**

1. what exactly is the baseline setting? not given in the current paper
2. what is the performance of the model performing one-step reasoning with "Prompt for first-try in step-level COT"?
3. what is the performance of the model performing overall retrieval of similar reasoning steps instead of stepwise?
4. The authors claim that only 30% more tokens are used, which is quite strange that even only retrieval for once would at least double the cost, so after all is it less than 30% of the examples need retrieval? if similar rejection techniques are applied to other retrieval baselines how would it work?

---

### Meta-Review · Area_Chair_n87z · 2025-12-06

**Summary:**

This paper proposes BoostStep, an approach integrating a step-aligned in-context learning mechanism to effectively enhance the math reasoning capability of LLMs.
The reviewers' concerns can be summarized as follows:
- Missing exeprimental details
- Outdated and selective benchmarking
- Dependency on example bank quality and step segmentation

**Reviewer Concerns:**

The concerns are still outstanding as the authors do not provide rebuttal.

**Reviewer Scores:**

The reviewers would keep their scores due to lack of rebuttal.

---

### Decision · Program_Chairs · 2026-01-26

Reject